# On quantum melting of superfluid vortex crystals: from Lifshitz scalar to dual gravity

Dung Xuan Nguyen[1,2] Sergej Moroz[3,4]

**1** Center for Theoretical Physics of Complex Systems, Institute for Basic Science(IBS), Daejeon, 34126, Korea
**2** Basic Science Program, Korea University of Science and Technology (UST), Daejeon 34113, Korea
**3** Department of Engineering and Physics, Karlstad University, Karlstad, Sweden
**4** Nordita, KTH Royal Institute of Technology and Stockholm University, Stockholm, Sweden

March 12, 2024

## Abstract

Despite a long history of studies of vortex crystals in rotating superfluids, their melting due to quantum fluctuations is poorly understood. Here we develop a fracton-elasticity duality to investigate a two-dimensional vortex lattice within the fast rotation regime, where the Lifshitz model of the collective Tkachenko mode serves as the leading-order low-energy effective theory. We incorporate topological defects and discuss several quantum melting scenarios triggered by their proliferation. Furthermore, we lay the groundwork for a dual non-linear emergent gravity description of the superfluid vortex crystals.

# 1 Introduction

Quantum vortices are topological defects in quantum superfluids which reveal quantum mechanics in these phases on macroscopic scales. Quantum vortex matter is an intriguing and multidisciplinary research field [1] which attracts both theorists and experimentalists. While being energetically costly excitations deep in the superfluid regime, condensation of vortices provides a natural framework for understanding of neighbouring non-supefluid phases and associated phase transitions [2–4].

In the superfluid regime vortices emerge in abundance at low temperatures provided the whole system is rotated [5–8]. As discovered first by Abrikosov [9] in a closely related context of type-II superconductors in an external magnetic field, in thermodynamic limit a regular vortex crystal ground state can emerge. It breaks spontaneously (magnetic) translation and rotation symmetries. In the two-dimensional limit, the study of low-energy collective excitations, known as Tkachenko waves [10], has been a subject of extensive theoretical investigation, as evidenced by works such as [11–22]. Additionally, an experimental observation of the Tkachenko waves have been successfully conducted at extremely low temperature in a cold atom experiment [23]. Notably, it was also suggested that the Tkachenko modes might explain the dynamics of pulsars [24].

Given that the two transverse Cartesian coordinates of a vortex constitute a canonical pair of variables [6, 25–27], it follows that vortices represent inherently fuzzy entities with an uncertainty area inversely proportional to the density of elementary bosons within the superfluid phase. Consequently, as the vortex density within the crystal approaches the magnitude of the boson density, quantum mechanical fluctuations in vortex positions become comparable to the distances between vortices. Rough estimates relying on the Lindemann criterion and small-scale exact diagonalization numerical simulations suggest that the vortex crystal experiences quantum melting at zero temperature when the filling fraction is roughly between 1 and 10 [6]. Here, the filling fraction, to be called $\nu$ in the following, is defined as the ratio between the boson density, $n_b$, and the vortex density, $n_v$. The precise nature of this quantum melting phenomenon remains poorly understood, representing a longstanding challenge in the field.

Fracton-elasticity duality [28–35] and its predecessors [36] provide an excellent framework to study possible melting mechanisms because it naturally incorporates disclinations and dislocations, which are topological defects in solids [37]. One can also easily incorporate vacancy and interstitial defects [29, 32]. In this formalism, quantum melting can be realized by a series of phase transitions, where dynamical defect fields play the role of the Higgs fields. This approach found practical application in the study of vortex crystals, as pioneered in [38]. In addition to computations of static interactions among various types of defects, this investigation uncovered several continuous quantum Higgs transitions triggered by condensation of the defects. Notably, it was found that the quantum melting of the vortex crystal might be preceeded by the condensation of vacancies or interstitials, leading to the emergence of an intermediate vortex supersolid phase, investigated originally in the classical finite-temperature problem [39, 40].

In this paper, we provide new insights into quantum melting of two-dimensional superfluid vortex crystals. Our starting point is the effective theory of the Tkachenko mode, which in the quadratic approximation reduces to a Lifshitz theory of a compact scalar field [19, 22, 40, 41]. This is a good coarse-grained description of the superfluid vortex crystal in a fast rotation limit, where the condensate occupies only the lowest Landau level. Within this field theory we discuss the fate of symmetry-allowed magnetic vertex operators that create vortex defects of the Lifshitz scalar that under special conditions correspond to vacancy and interstitial defects in the vortex crystal. Taking inspiration from the previous work [3, 42], we determine at which filling $\nu$ such magnetic vertex operators are relevant in the renormalization group (RG) sense and thus destabilize the Lifshitz description of the vortex crystal. This sheds some new light on the vortex supersolid

regime discussed in [38].

Recent surge of interest in fracton models inspired the authors of [43,44] to develop fractonic gauge duals of various realizations of the compact Lifshitz theory in two spatial dimensions. Here, we develop a simple and elegant quadratic traceless symmetric tensor gauge theory that is dual to the Lifshitz scalar description of the superfluid vortex crystal. Within this framework we investigate the crystalline and fluid phases and study corresponding topological defects and excitation modes. Using [45], we also consider and speculate about an exotic direct quantum phase transition between the vortex solid and fluid.

While capturing the quadratic dispersion of the low-lying Tkachenko mode, the quadratic Lifshitz theory has an important shortcoming in describing the superfluid vortex crystal as it does not realize a non-commutative algebra of magnetic translation symmetries. Recently, a non-linear non-commutative field theory, which reduces to the Lifshitz model in the quadratic approximation, was proposed that incorporates all physical symmetries [22]. This theory was used to determine the decay rate of the Tkachenko quanta at low energies. Here, starting from the linearized fractonic duality of the Lifshitz theory of the vortex crystal, we make first steps towards a non-linear dual of the non-linear theory [22]. We argue that this dual must be a dynamical theory of bimetric gravity and identify some of its gauge-invariant building blocks.

## 2 Lifshitz effective theory of vortex crystal

### 2.1 Tkachenko modes

Low-energy excitations of a two-dimensional vortex crystal in a rotating superfluid emerge from intertwined superfluid and elastic couarse-grained fluctuations. Employing the boson-vortex duality [46,47], the leading-order quadratic effective theory [20] in the lowest Landau level approximation[1] is given by the following Lagrangian [21]

$$\mathcal{L}^{(2)} = -\frac{Bn_0}{2}\epsilon_{ij}u^i\dot{u}^j + Be_iu^i - \frac{\lambda}{2}b^2 - \mathcal{E}_{\text{el}}\left(u_{ij}\right). \quad (2.1)$$

Here the building blocks are the coarse-grained crystal displacement field $u^i$ and a dual $u(1)$ gauge field $a_\mu$. In spirit of the boson-vortex duality, the superfluid density fluctuations $\delta n = n - n_0$ and superfluid current $j^i$ are fixed by the dual magnetic field $b = \epsilon^{ij}\partial_i a_j$ and the electric field $e_i = \partial_t a_i - \partial_i a_0$, respectively. The first term in the Lagrangian (2.1), encodes the Berry phase of vortices moving in the superfluid. Here $B$ denotes an effective magnetic field experienced by elementary bosons due to external rotation and $n_0$ is the average superfluid density. The second term in Eq. (2.1) represents an effective dipole energy acquired by vortices away from their elastic equilibrium. The superfluid internal energy is a function of the superfluid density and in Eq. (2.1) it was expanded around the ground state value $n_0$ to quadratic order in the density fluctuations $b = \delta n$. Finally, for a two-dimensional triangular vortex crystal, the elastic energy density $\mathcal{E}_{\text{el}}\left(u_{ij}\right) = 2C_1u_{kk}^2 + 2C_2\tilde{u}_{ij}^2$ with $\tilde{u}_{ij} = u_{ij} - \left(u_{kk}\delta_{ij}\right)/2$ being the traceless part of the symmetric strain tensor $u_{ij} = \left(\partial_i u_j + \partial_j u_i\right)/2$. The coefficients $C_1$ and $C_2$ are the compression and shear elastic moduli, respectively.

The $u(1)$ Gauss law $\partial_i u^i = 0$ implies immediately that the two components of the dispacement field are not independent. To hardwire the transverse nature of displacements, we can introduce a dimensionless scalar field $\phi$ such that $u^i = \tilde{\partial}^i\phi/B$, where we introduced a skew derivative

---

[1]To go beyond the lowest Landau level approximation, one should add a kinetic superfluid contribution that in the dual description is represented by a subleading electric term $\sim m\mathbf{e}^2/(2n_0)$, where $m$ denotes the mass of the elementary boson particles. This addition gives rise to the celebrated gapped Kohn mode, but does not modify the quadratic Tkachenko dispersion (2.3) at low momenta [20].

$\tilde{\partial}^i = \epsilon^{ij}\partial_j$. In addition to parametrizing allowed transverese displacements, the field $\phi$ also represents a coarse-grained $2\pi$-periodic superfluid phase [22].

Integrating now out the dual $u(1)$ gauge field, one arrives at the quantum Lifshitz model representation of the vortex crystal [19, 22, 40, 41]

$$
\begin{aligned}
\mathcal{L}_\phi &= \frac{1}{2\lambda}\dot{\phi}^2 - 2C_2\tilde{u}_{ij}^2 \\
&= \frac{1}{2\lambda}\dot{\phi}^2 - \frac{C_2}{2B^2}(\partial_i\tilde{\partial}_j\phi + \partial_j\tilde{\partial}_i\phi)^2
\end{aligned}
\tag{2.2}
$$

which encodes the low-energy transverse Tkachenko excitations with a quadratic dispersion relation

$$
\omega^2 = \frac{2C_2\lambda}{B^2}q^4.
\tag{2.3}
$$

This agrees with the known low-momentum limit of the collective Tkachenko excitation of the vortex crystal in the lowest Landau level approximation [16, 48, 49], where the shear elastic modulus $C_2$ is known to be [16, 48]

$$
C_2 = 0.119\lambda n_0^2.
\tag{2.4}
$$

Although the low-momentum limit of the Tkachenko dispersion is encoded properly in the quadratic Lifshitz model, this theory does not capture a non-commutative algebra of magnetic translation symmetries. The theory (2.2) is only a quadratic truncation of a non-linear non-commutative Goldstone theory [22] that respects all physical symmetries of the problem.

Remarkably, due to the transverse nature of the allowed displacements, the time-reversal breaking Berry term from Eq. (2.1) completely drops out from the Lifshitz model (2.2). Note, however, that in the presence of topological vortex configurations in the field $\phi$, after integration by parts the Berry term survives at the cores of those defects. Physically, vortices of the Tkachenko field represent vacancies and interstitials in the vortex crystal. The resulting contribution to the action is given by

$$
\mathcal{S}_B = -\frac{\nu}{2}\int dt d^2x(j_v^t\partial_t\phi - j_v^i\partial_i\phi),
\tag{2.5}
$$

where we introduced the defect three-current $j_v^\mu = \frac{1}{2\pi}\epsilon^{\mu\nu\rho}\partial_\nu\partial_\rho\phi$ which encodes both the density and spatial current carried by the vortices of the Tkachenko field $\phi$.

## 2.2 Condensation of vacancies and interstitials

Here within the quantum Lifshitz theory, we investigate proliferation of vacancies and interstitials in a two-dimensional superfluid vortex crystal at vanishing temperature.[2] Microscopically, we have in mind the supersolid scenario by Andreev and Lifshitz [50]: In the crystal, isolated vacancies and interstitials cost finite energy to create, but due to quantum tunneling, the bottom of their energy band touches zero and they become gapless. Provided this happens (which should be verified in a microscopic calculation), here we want to clarify if such condensation of vacancies/interstitials is an RG-relevant perturbation that destabilizes the vortex crystal phase captured by the quantum Lifshitz model (2.2).

Up to surface terms, the effective theory (2.2) of the compact scalar $\phi \in (0, 2\pi)$ is equivalent to the $z = 2$ Lifshitz theory

$$
\mathcal{L} = \frac{1}{2\lambda}\dot{\phi}^2 - \frac{C_2}{B^2}(\Delta\phi)^2.
\tag{2.6}
$$

---

[2]At finite temperature superconductors this problem was discussed in detail in an Abrikosov crystal in [39].

We now rescale the Tkachenko field $\varphi = \sqrt{\lambda}\phi$, so the Lagrangian takes the canonical form

$$\mathcal{L} = \frac{1}{2}\dot{\varphi}^2 - \frac{\eta^2}{2}(\Delta\varphi)^2 \tag{2.7}$$

with $\eta^2 = 2C_2\lambda/B^2$. Using now Eq. (2.4), one finds $\eta \approx 0.488\lambda n_0/B$. Notice that the field rescaling implies that $\varphi \in (0, 2\pi/\sqrt{\lambda})$.

We are interested in operators that create vacancies and interstitials in the vortex crystals. In the Lifshitz effective theory they correspond to magnetic vertex operators [42] that create vortex defects of the field $\varphi$. To create such a vortex centered around a position $\mathbf{x}$, one should act on the vortex crystal vacuum with

$$\tilde{O}_{\tilde{m}}(\mathbf{x}) = \exp(i \int d^2\mathbf{z}\,\alpha(\mathbf{z})\Pi(\mathbf{z})), \tag{2.8}$$

where $\alpha(\mathbf{z}) = \tilde{m}\arg(\mathbf{z} - \mathbf{x})$ and $\Pi(\mathbf{z})$ denotes a canonical momentum density that is conjugate to the Lifshitz field $\varphi$. To account for the rescaled radius of the redefined field $\varphi$, here $\tilde{m} = m/\sqrt{\lambda}$ with $m$ an integer. Elementary vacancies and interstitials correspond to $m = \pm 1$. The magnetic vertex operators can be added to the Lagrangian of the vortex lattice since they do not break any global symmetry. Noteably, in the quatum Lifshitz theory the static correlation functions of the vertex operators are known exactly [3, 42] and are fixed by the parameter $\eta$, see Appendix A. This allows to extract scaling dimensions of the vertex operators. For an elementary vortex ($m = \pm 1$) of $\phi$ that corresponds to an elementary vacancy/interstitial we find

$$\Delta_v = 0.488\frac{n_0}{B/(2\pi)} = 0.488\nu, \tag{2.9}$$

where the filling fraction $\nu = n_b/n_v$ with the vortex density $n_v = B_0/(2\pi)$. We observe that at large filling fraction ($\nu \to \infty$), where the system is deep in the Gross-Pitaevskii vortex lattice regime, $\Delta_v \gg 1$ and vacancies/interstitials are irrelevant. However, as $\nu$ decreases (and the shear modulus $C_2$ softens), the vacancy turns marginal $\Delta_v = 2 + z = 4$ at the critical filling $\nu_c \approx 4/0.488 \approx 8.2$.[3]

Now in a given microscopic model, if a vortex vacancy/interstitial becomes gapless at a filling $\nu > \nu_c$, the perturbaton is RG-irrelevant and one expects that the Lifshitz fixed point is stable. On the other hand, if the defect becomes gapless at $\nu < \nu_c$, it will distabilize the Lifshitz fixed point because it is an RG-relevant perturbation. The detailed investigation of such instability is left to a future work, but we anticipate that it might shed new light on the mysterious vortex supersolid regime [32, 38–40].

## 3 Dual tensor gauge theory

### 3.1 Vortex crystal

Given that the low-momentum strain of the Tkachenko excitations is symmetric and traceless, we will dualize the Lifshitz theory (2.2) to a symmetric traceless gauge theory coupled to a scalar charge [51]. To this end, first introduce the Hubbard-Stratonovich fields $b$ and $e^{ij}$

$$\mathcal{L} = \frac{\kappa}{8}e_{ij}e^{ij} - \frac{\lambda}{2}b^2 - \frac{1}{2B}e^{ij}(\partial_i\tilde{\partial}_j\phi + \partial_j\tilde{\partial}_i\phi) + b\partial_t\phi \tag{3.1}$$

---

[3]Notice that in this paper, following [3, 42], the bare Lifshitz theory (2.6) was used to calculate the vacancy/interstitial correlation function. The incorporation of the Berry term (2.5) into this calculation is left to a future work.

with $\kappa = C_2^{-1}$ and $e^{ij}$ being a symmetric and traceless tensor. Solving the equations of motion, one finds, $b = \partial_t \phi / \lambda$ and $e_{ij} = 2(\partial_i \tilde{\partial}_j \phi + \partial_j \tilde{\partial}_i \phi)/(\kappa B)$. The field $b$ represents fluctuations of the coarse-grained superfluid density $n$ in the vortex crystal [22]. On the other hand, $e^{ij}$, as the variation of the action with respect to the shear strain, is the traceless part of the stress tensor $T^{ij}$.

Now we split the Lifshitz field $\phi = \phi^{(r)} + \phi^{(s)}$, where the regular part $\phi^{(r)}$ is smooth, while the singular part $\phi^{(s)}$ contains contributions from topological defects (disclinations and dislocations) of the vortex crystal. For now, we will assume that the phase field $\phi$ has no vortex singularities, i.e. $\epsilon^{\mu\nu\rho} \partial_\nu \partial_\rho \phi = 0$, but only higher derivative singularites that correspond to disclinations and dislocations. In other words, there are no vacancies and interstitials at low energies, which justifies why the compression part of the elastic energy is dropped in our departure point (2.2).

Integrating out the regular part $\phi^{(r)}$, we find the conservation law

$$\partial_t b + \frac{1}{2B}(\partial_i \tilde{\partial}_j + \partial_j \tilde{\partial}_i) e^{ij} = 0. \tag{3.2}$$

This equation has a simple physical interpretation as a consequence of the lowest Landau level limit of the momentum and particle number conservations [52]. In this limit, due to the absence of inertia, the particle number current is fixed by equating the Lorentz force and the exerted stress which gives $j_i = -\epsilon_{ij} \partial_k T^{jk}/B$. As a result, Eq. (3.2) is just the particle number conservation equation when restricted to the lowest Landau level [52, 53]

$$\partial_t n + \frac{1}{B}\tilde{\partial}_i \partial_j T^{ij} = 0. \tag{3.3}$$

By introducing a traceless symmetric gauge potential $a_{ij}$ and representing the magnetic field $b$ and the traceless symmetric electric field $e_{ij}$ as

$$b = -\frac{1}{2B}(\partial_i \tilde{\partial}_j + \partial_j \tilde{\partial}_i) a_{ij}, \tag{3.4}$$

$$e_{ij} = \partial_t a_{ij} - (\partial_i \partial_j - \frac{1}{2}\delta_{ij}\Delta) a_0, \tag{3.5}$$

the conservation law (3.2) becomes the Bianchi identity of a symmetric traceless tensor gauge theory. Both $b$ and $e_{ij}$ are invariant under the following $u(1)$ gauge transformation

$$a_0 \to a_0 + \partial_t \beta, \qquad a_{ij} \to a_{ij} + (\partial_i \partial_j - \frac{1}{2}\delta_{ij}\Delta)\beta \tag{3.6}$$

which preserves the traceless form of the gauge potential $a_{ij}$.

The gauge theory encodes only one physical degree of freedom because the two components of the traceless symmetric tensor $a_{ij}$ are reduced to one due to the $u(1)$ gauge redundancy. The excitation mode of the dual tensor gauge theory

$$\mathcal{L} = \frac{\kappa}{8} e_{ij} e^{ij} - \frac{\lambda}{2} b^2 \tag{3.7}$$

has the quadratic dispersion (2.3), see Appendix B. As we demonstrate in Appendix C, the simple gauge theory (3.7) can be obtained from a more complicated dual theory with intertwined tensor (dual to elasticity) and vector (dual to superfluidity) gauge fields that was derived and analysed in Ref. [38].

The field equation for $a_0$ is the Gauss law of the gauge theory that constraints the stress tensor as $\partial_i \partial_j e_{ij} = 0$.

The topological defects of the vortex lattice encoded in the singular part $\phi^{(s)}$ of the Tkachenko field couple naturally to the tensor gauge theory. Using integration by parts, we end up with the following result

$$\mathcal{L} = \frac{\kappa}{8} e_{ij} e^{ij} - \frac{\lambda}{2} b^2 - \rho a_0 + j^{ij} a_{ij} \tag{3.8}$$

with the isolated disclination charge density [28, 37]

$$\rho = \frac{1}{2B}(\tilde{\partial}_j\tilde{\partial}_i - \frac{1}{2}\delta_{ij}\Delta)[\partial_i\tilde{\partial}_j + \partial_j\tilde{\partial}_i]\phi^{(s)} = \frac{1}{2B}\tilde{\partial}_j\tilde{\partial}_i[\partial_i\tilde{\partial}_j + \partial_j\tilde{\partial}_i]\phi^{(s)} = \tilde{\partial}_j\tilde{\partial}_i u_{ij}^{(s)} \qquad (3.9)$$

and the symmetric tensor current

$$j^{ij} = \frac{1}{2B}\left(\partial_t[\partial_j\tilde{\partial}_i + \partial_i\tilde{\partial}_j] - [\partial_j\tilde{\partial}_i + \partial_i\tilde{\partial}_j]\partial_t\right)\phi^{(s)}. \qquad (3.10)$$

With the assumption of vanishing vacancies $\epsilon^{\mu\nu\lambda}\partial_\nu\partial_\lambda\phi^{(s)} = 0$, and in addition $\epsilon^{\mu\nu\lambda}\partial_\nu\partial_\lambda\partial_t\phi^{(s)} = 0$, one can show that

$$j^{ij} \to \frac{1}{2}\left[(\partial_t\partial_i - \partial_i\partial_t)\,u_j^{(s)} + (\partial_t\partial_j - \partial_j\partial_t)\,u_i^{(s)}\right], \qquad (3.11)$$

which is simply related to the conventional dislocation current $J^{ij} = \epsilon^{ik}\epsilon^{jl}j_{kl}$ [37].

From the gauge symmetry (3.6), the defect density and current satisfy the conservation law

$$\partial_t\rho + (\partial_j\partial_i - \frac{1}{2}\delta_{ij}\Delta)j^{ij} = 0. \qquad (3.12)$$

The equation (3.12) implies conservation of particles, dipoles and the trace of the quadrupoles[4]

$$Q = \int d^2x\rho, \qquad Q^i = \int d^2x\epsilon^{ij}x^j\rho, \qquad Q^{tr} = \int d^2x\mathbf{x}^2\rho. \qquad (3.13)$$

As a result, isolated gauge charges are immobile, gauge dipoles are conserved and can only move perpendicular to their dipole moment, while gauge quadrupoles are free to move. We thus identify charges with lattice disclinations and dipoles with lattice dislocatioins which can glide along their Burgers vector, but cannot climb. Mathematically, they satisfy the glide constraint [5]

$$\delta_{ij}j^{ij} = \delta_{ij}J^{ij} = 0. \qquad (3.14)$$

It is now straightforward to compute a static interaction potential between disclinations. Integrating out the gauge field $a_0$, one finds

$$\mathcal{L} = -\frac{1}{2}\rho(-q)\frac{8C_2}{q^4}\rho(q). \qquad (3.15)$$

In real space it gives rise to a harmonic attractive potential. As a result, in the vortex crystal disclinations are very costly in energy. They usually do not appear in isolation, but are bound together into dislocations.

To determine the interaction potential between dislocations, we consider charge density $\rho$ induced by dipoles, i.e. $\rho = \epsilon_{ij}\partial_i\chi_j$, where we introduced the Burgers vector density $\chi_i = \epsilon^{ab}\partial_a\partial_b u_i^{(s)}$ [37]. In momentum space, we can rewrite Eq. (3.15) as

$$\mathcal{L} = -\frac{1}{2}\chi_i^T(-q)\frac{8C_2}{q^2}\chi_i^T(q), \qquad (3.16)$$

where we introduced the transverse projection $\chi_i^T(q) = \left(\delta^{ij} - q^iq^j/q^2\right)\chi_j(q)$. This agrees with the lowest Landau level limit of the previous result [38, 54]. We observe that dislocation interact via an anisotropic long-range logarithmic potential.

---

[4]These three conservation laws also follow from the Gauss law.

[5]One can derive this equation directly using equation (3.10) assuming no vortex singularities in $\phi^{(s)}$ and $\partial_t\phi^{(s)}$.

In the presence of vacancies and interstitials, the construction of the dual gauge theory must be modified because (i) the stress tensor is not traceless anymore, (ii) the Berry term (2.5) is present. Moreover, the defect current $j^{ij}$ is not traceless either. From Eq. (3.10) its trace is given by

$$\delta_{ij}j^{ij} = \frac{2\pi}{B}\partial_t j_v^t, \tag{3.17}$$

where we used the vacancy density $j_v^t = \frac{1}{2\pi}\epsilon^{ij}\partial_i\partial_j\phi$. In addition, in presence of a vacancy current $j_v^k$, the relation between the tensor current $j^{ij}$ (3.10) and the dislocation current $J^{ij}$ [37] takes the form

$$J^{ij} = \epsilon^{ik}\epsilon^{jl}\left[j^{kl} + \frac{\pi}{B}\left(\partial_k j_v^l + \partial_l j_v^k\right)\right]. \tag{3.18}$$

Combining now the last two equations, we arrive at the modified glide constraint[6]

$$\delta_{ij}J^{ij} - 2\pi B\partial_\mu j_v^\mu = 0. \tag{3.19}$$

Now dislocations can climb at expense of creating or destroying vortex vacancies resulting in the modification of the conserved charge $Q^{tr}$ in Eq. (3.12) to

$$\tilde{Q}^{tr} = \int d^2x \left(\mathbf{x}^2\rho - \frac{4\pi}{B}j_v^t\right), \tag{3.20}$$

see Appendix C. In the vortex crystal, vacancies interact via a short-range potential whose nature depends on the interplay of the compression and shear elastic moduli [38]. Note that in order to fix the interaction constant, one needs to go beyond our theory (2.2) which for example misses elastic compression contributions to the interaction potential between vacancies.

## 3.2 Vortex fluid

In this section we propose a simple field theory of a fully gapped vortex fluid phase where all global symmetries of the ground state are restored. To this end, consider a Lifshitz theory of a compact scalar $\chi$ minimally coupled to the $u(1)$ traceless symmetric tensor gauge theory derived in the previous section. Physically, we can think of the scalar $\chi$ as representing the phase of a (complex) disclination field that serves as the Higgs field in the vortex fluid phase. Its Lagrangian is

$$\mathcal{L}_\chi = \frac{\tau}{2}\underbrace{(\partial_t\chi - a_0)}_{\mathcal{D}_t\chi}{}^2 - \frac{\sigma}{2}\left(\underbrace{[\partial_i\partial_j - \frac{1}{2}\delta_{ij}\Delta]\chi - a_{ij}}_{\mathcal{D}_{ij}\chi}\right)^2. \tag{3.21}$$

Under $u(1)$ gauge transformations $\chi \to \chi + \beta$, so the covariant derivatives $\mathcal{D}_t\chi$ and $\mathcal{D}_{ij}\chi$ are gauge invariant.

The corresponding field equation for $\chi$ is exactly the conservation law (3.12)

$$\partial_t\underbrace{(\tau\mathcal{D}_t\chi)}_{\rho} + (\partial_i\partial_j - \frac{1}{2}\delta_{ij}\Delta)\underbrace{(\sigma\mathcal{D}_{ij}\chi)}_{j^{ij}} = 0, \tag{3.22}$$

where $\rho$ and $j^{ij}$ is the disclination density and symmetric tensor current. As a result the charges $Q$, $Q^i$ and $Q^{tr}$ introduced in Eq. (3.13) are all automatically conserved. In the unitary gauge $\mathcal{D}_t\chi \to -a_0$ and $\mathcal{D}_{ij}\chi \to -a_{ij}$ and in that particular gauge the conservation law is

$$\tau\partial_t a_0 + \sigma\partial_i\partial_j a_{ij} = 0, \tag{3.23}$$

---

[6]We note that the definition of the vacancy current $j_v^\mu = 1/(2\pi)\epsilon^{\mu\nu\rho}\partial_\nu\partial_\rho\phi$ used in this paper differs by the factor $1/(2\pi)$ from the definition used in Ref. [38].

where we used that $a_{ij}$ is traceless.

Now we compute dispersion relations of excitation modes of the gauge theory coupled to the Lifshitz matter. We start from the complete Lagrangian

$$\mathcal{L} = \frac{\kappa}{8} e_{ij} e^{ij} - \frac{\lambda}{2} b^2 + \frac{\tau}{2} (\mathcal{D}_t \chi)^2 - \frac{\sigma}{2} (\mathcal{D}_{ij} \chi)^2. \tag{3.24}$$

The corresponding Gauss law

$$\frac{\kappa}{4} \partial_i \partial_j e_{ij} + \rho = 0, \tag{3.25}$$

while the Ampere law $\frac{\delta S}{\delta a_{ij}} = 0$ is

$$\frac{\kappa}{4} \partial_t e_{ij} - \frac{\lambda}{2B} (\partial_i \tilde{\partial}_j + \partial_j \tilde{\partial}_i) b = j^{ij}. \tag{3.26}$$

To solve it, we first rewrite this equation in terms of the gauge potentials $a_0$ and $a_{ij}$. Working in the unitary gauge and using Eq. (3.23) allows to eliminate completely the scalar potential $a_0$. So one ends up with the equation for $a_{ij}$

$$\frac{\kappa}{4} \left( \partial_t^2 a_{ij} + \frac{\sigma}{\tau} (\partial_i \partial_j - \frac{1}{2} \delta_{ij} \Delta) \partial_k \partial_l a_{kl} \right) + \frac{\lambda}{4B^2} (\partial_i \tilde{\partial}_j + \partial_j \tilde{\partial}_i)(\partial_k \tilde{\partial}_l + \partial_l \tilde{\partial}_k) a_{kl} + \sigma a_{ij} = 0. \tag{3.27}$$

To simplify the calculation of the dispersion relation, we will use isotropy and consider a mode propagating in the $x$-direction. For the trace components $a_{11} = -a_{22} = f$, Eq. (3.27) simplifies to

$$\frac{\kappa}{4} [\partial_t^2 + \frac{\sigma}{2\tau} \partial_x^4] f + \sigma f = 0 \tag{3.28}$$

which leads to the gapped dispersion of the $f$-mode

$$\omega^2 = \frac{2\sigma}{\kappa\tau} q^4 + \frac{4\sigma}{\kappa}. \tag{3.29}$$

For the off-diagonal components $a_{12} = a_{21} = g$, we find

$$\frac{\kappa}{4} [\partial_t^2 + \frac{\lambda}{2B^2} \partial_x^4] g + \sigma g = 0. \tag{3.30}$$

We observe that the $g$-mode (which corresponds to the gapless Tkachenko mode in the absence of the Lifshitz matter, see Appendix B) acquires a gap due to the coupling to the Lifshitz matter sector

$$\omega^2 = \frac{2\lambda}{\kappa B^2} q^4 + \frac{4\sigma}{\kappa}. \tag{3.31}$$

We thus end up with two physical modes that have the same gap $\Delta^2 = 4\sigma/\kappa$ at $q = 0$. In spirit, these results resemble physical excitations in a superconductor, where the longitudinal and transverse excitation modes have the same energy gap [55].

Similarly to superconductivity, the vortex fluid exhibits a (dual) Meissner effect. Specifically, in the static limit $\omega = 0$, from the dispersion (3.31) we find $q = e^{i\pi/2}/\lambda_L$, where the dual London penetration length is $\lambda_L = \sqrt[4]{2\sigma B^2/\lambda}$. As the result, the dual magnetic field $b \sim \partial_i \tilde{\partial}_j a_{ij}$ which represents fluctuations of the superfluid density, near the boundary of the system decays exponentially into the bulk.

In summary, the $u(1)$ tensor gauge theory coupled to the Lifshitz matter represents a fully gapped vortex fluid phase, where global symmetries (magnetic translations and rotations) are respected by the ground state. Being produced by the dual Higgs mechanism, this phase has many properties similar to $u(1)$ superconductors.

## 3.3 Ginzburg-Landau theory and the Higgs transition

One might wonder if the vortex crystal from Sec. 3.1 can undergo a direct quantum melting transition to the isotropic vortex fluid from Sec. 3.2. Here we write down a simple Ginzburg-Landau theory that achieves that.

We consider a complex scalar $\Psi$ that represents a disclination annihilation and plays the role of the Higgs field. Under $u(1)$ gauge transformations it is postulated to transform as

$$\Psi \to e^{ir\beta}\Psi, \tag{3.32}$$

where $r$ is the $u(1)$ gauge charge of the Higgs field $\Psi$. We define the covariant derivative

$$D_{ij}\Psi^2 = \Psi\partial_i\partial_j\Psi - \partial_i\Psi\partial_j\Psi - \frac{1}{2}\delta_{ij}\left(\Psi\Delta\Psi - \partial_k\Psi\partial^k\Psi\right) - ira_{ij}\Psi\Psi, \tag{3.33}$$

that transforms covariantly

$$D_{ij}\Psi^2 \to e^{i2r\beta}D_{ij}\Psi^2 \tag{3.34}$$

under the gauge transformations (3.6) and (3.32). This covariant derivative is the traceless symmetric version of the one considered in Ref. [45]. Moreover, we can define the temporal covariant derivative

$$D_t\Psi = \partial_t\Psi - ira_0 \tag{3.35}$$

that also transforms covariantly

$$D_t\Psi \to e^{ir\beta}D_t\Psi. \tag{3.36}$$

From these building blocks, we now can write down the following Ginzburg-Landau Lagrangian

$$\mathcal{L}_\Psi = \frac{i}{2}\Psi^\dagger D_t\Psi - \frac{m}{2}|D_{ij}\Psi^2|^2 - v_2\Psi^\dagger\Psi - \frac{v_4}{2}|\Psi^\dagger\Psi|^2. \tag{3.37}$$

Notice that in contrast to the ordinary Ginzburg-Landau theory, the term with spatial derivatives is quartic in $\Psi$ and thus represents interactions. The gauged dipole symmetry $Q^i$ from Eq. (3.13) prohibits quadratic terms in $\Psi$ with spatial derivatives, while the quadrupole conservation $Q^{tr}$ imposes the traceless condition for the covariant derivative (3.33).

We can now use the parameter $v_2$ to tune between the two phases. When $v_2 < 0$, the theory is in the Higgs phase. We write $\Psi = \sqrt{\psi}e^{ir\chi}$ with $\psi = |v_2|/v_4 + \gamma$, where $\gamma$ is a radial massive fluctuation. After integrating out $\gamma$ and keeping the leading order terms, we obtain

$$\mathcal{L}_\chi = \frac{r^2}{2v_4}\left(\partial_t\chi - a_0\right)^2 - \frac{mr^2v_2^2}{2v_4^2}\left([\partial_i\partial_j - \frac{1}{2}\delta_{ij}\Delta]\chi - a_{ij}\right)^2. \tag{3.38}$$

After renaming

$$r^2/v_4 \to \tau, \quad \frac{mr^2v_2^2}{v_4^2} \to \sigma, \tag{3.39}$$

we recover the effective theory of the vortex fluid, i.e., the Lifshitz scalar coupled to the dual tensor gauge theory (3.21).[7]

On the other hand, for $v_2 > 0$, the Higgs field $\Psi$ is gaped and at low energies it decouples from the gauge theory (3.7). We are thus in the vortex crystal phase that supports the quadratically dispersing Tkachenko mode.

---

[7]One can consider a different Ginzburg-Landau Lagrangian

$$\mathcal{L}'_\Psi = \frac{1}{2}|D_t\Psi|^2 - \frac{m}{2}|D_{ij}\Psi^2|^2 - v_2\Psi^\dagger\Psi - \frac{v_4}{2}|\Psi^\dagger\Psi|^2. \tag{3.40}$$

After integrating out the massive fluctuation $\gamma$, one arrives at a similar quadratic form of the Golstone boson action in the Higgs phase but with different coefficients.

In the mean-field approximation, the quantum transition at $v_2 = 0$ between the vortex crystal and vortex fluid is direct and continuous. Of course, this simple picture might not survive quantum fluctuations near $v_2 = 0$ that could lead to split transitions, as discussed for example in [38,43]. A careful treatment of this problem is left to a future work.

## 4 Towards a dual gravitational theory

The symmetric tensor gauge field $a_{ij}$ is reminiscent of a metric in a (linearized) theory of gravity. Indeed, already Kleinert noticed the analogy between the tensor gauge formulation of elasticity and Einstein's theory of gravity, and suggested that gravity could emerge from the defects of a crystal with lattice spacing of order the Plank length [56,57]. Later, the symmetric tensor description of a boson liquid phase was also proposed as a gravity theory in works by Xu and collaborators [58,59]. Pretko also formulated the fractonic symmetric tensor gauge theory in terms of a gravity model with both positive and negative mass matter fields [60].

In this paper we follow closely [53]. To construct the dynamical gravity from the dual tensor gauge theory, we first introduce a symmetric and traceless dimensionless field

$$\mathfrak{h}_{ij} = -l^2(\varepsilon_{ik}a_{jk} + \varepsilon_{jk}a_{ik}) \tag{4.1}$$

that contains equivalent information to $a_{ij}$. Here $l = 1/\sqrt{B}$ is the magnetic length. Under $u(1)$ gauge transformations (3.6), $\mathfrak{h}_{ij}$ transforms as

$$\mathfrak{h}_{ij} \to \mathfrak{h}_{ij} - l^2(\partial_j \tilde{\partial}_i + \partial_i \tilde{\partial}_j)\beta \tag{4.2}$$

which can be rewritten as

$$\mathfrak{h}_{ij} \to \mathfrak{h}_{ij} - \partial_i \xi_j - \partial_j \xi_i, \tag{4.3}$$

where we introduced $\xi^i = l^2 \tilde{\partial}^i \beta$ that by construction satisfies $\partial_i \xi^i = 0$. We recognize the linearized transformation of a metric tensor fluctuation under volume-preserving diffeomorphisms.

Now we are ready to extend to a non-linear realization of the $u(1)$ gauge redundancy. To that end, we introduce a dynamical unimodular metric $\mathfrak{g}_{ij}$ that under volume-preserving infinitesimal diffeomorphisms $x^i \to x^i + \xi^i = x^i + \ell^2 \varepsilon^{ij} \partial_j \beta$ transforms as

$$\delta_\beta \mathfrak{g}_{ij} = -\xi^k \partial_k \mathfrak{g}_{ij} - \mathfrak{g}_{kj} \partial_i \xi^k - \mathfrak{g}_{ik} \partial_j \xi^k = -\ell^2 \varepsilon^{kl} \left( \partial_k \mathfrak{g}_{ij} + \mathfrak{g}_{kj} \partial_i + \mathfrak{g}_{ik} \partial_j \right) \partial_l \beta. \tag{4.4}$$

In the linear regime, where $\mathfrak{g}_{ij} = \delta_{ij} + \mathfrak{h}_{ij} + O\left(\mathfrak{h}^2\right)$, we recover the transformation (4.3). Notice that we are dealing with a bimetric theory[8] because in additional to the dual dynamical metric $\mathfrak{g}_{ij}$, we have at our disposal also a background symmetric metric tensor $g_{ij}$ that measures distances between elementary bosons on the surface, where the vortex crystal is formed. To have a periodic crystal structure, this background metric is assumed to be flat.

Following [53], the non-linear generalization of the $u(1)$ gauge transformations of the field $a_0$ that satisfies the non-commutative algebra of volume-preserving diffeomorphisms $[\delta_\alpha, \delta_\beta] = \delta_{[\alpha,\beta]}$ is

$$\delta_\beta a_0 = \partial_t \beta - \xi^k \partial_k a_0 = \partial_t \beta - \ell^2 \varepsilon^{kl} \partial_k a_0 \partial_l \beta. \tag{4.5}$$

Given these objects and their transformation properties, we will look now for the basic building blocks of the non-linear dual gravity theory that correspond to the magnetic and electric fields of the traceless tensor gauge theory (3.7).

---

[8]A gapped bimetric theory of fractional quantum Hall fluids was proposed and analyzed in [61].

From the dynamical metric $\mathfrak{g}_{ij}$ (and its inverse $\mathfrak{g}^{ij}$), we can construct the gauge-invariant Ricci scalar $\mathfrak{R}$ [62] that in two spatial dimensions fixes completely the Riemann tensor.[9] In the linearized regime

$$\mathfrak{R} = \partial_i \partial_j \mathfrak{h}_{ij} = -2b. \tag{4.6}$$

We thus find that in the non-linear realization, the Ricci scalar $\mathfrak{R}$ corresponds to the magnetic field $b$ which represents fluctuations of the coarse-grained superfluid density in the vortex crystal.

Although the time-derivative of the dynamical metric does not transform nicely under time-dependent gauge transformations, we can introduce a traceless "shear strain rate" tensor [63, 64]

$$\mathfrak{s}_{ij} = \partial_t \mathfrak{g}_{ij} + \nabla_i v_j + \nabla_j v_i - \mathfrak{g}_{ij} \nabla_k v^k, \tag{4.7}$$

where the covariant derivatives were defined using the dynamical metric $\mathfrak{g}_{ij}$. Here we also introduced the velocity vector field $v^i = l^2 \varepsilon^{ij} \partial_j a_0$ that under volume-preserving diffeomorphisms transforms as

$$\delta_\beta v^i = -\xi^k \partial_k v^i + v^k \partial_k \xi^i + \dot{\xi}^i. \tag{4.8}$$

Physically, up to epsilon contractions, the tensor $\mathfrak{s}_{ij}$ represents the traceless part of the physical stress tensor. In the linearized regime it thus essentially reduces to to the electric field $e_{ij}$.

Since the Einstein-Hilbert integral $\int d^2x \sqrt{\mathfrak{g}} \mathfrak{R}$ is fixed by the topological Euler characteristic $\chi_E$ of a two-dimensional manifold, the first non-trivial term including the Ricci scalar is $\sim \mathfrak{R}^2$. We can also raise the indices $\mathfrak{s}^{ij} = \mathfrak{g}^{ik} \mathfrak{g}^{jl} \mathfrak{s}_{kl}$ and construct a dynamical scalar contribution $\sim \mathfrak{s}_{ij} \mathfrak{s}^{ij}$ to the non-linear theory. The combination of the two terms

$$\mathcal{L} = \frac{\kappa}{32\ell^4} \mathfrak{s}_{ij} \mathfrak{s}^{ij} - \frac{\lambda}{8} \mathfrak{R}^2 \tag{4.9}$$

reduces in the linearized effective theory (3.7).

Using the dynamical metric tensor $\mathfrak{g}_{ij}$ and the "shear strain rate" tensor $\mathfrak{s}_{ij}$, we can construct a more general non-linear gravity action that is invariant under the non-commutative volume-preserving diffeomorphism transformations (4.4), (4.5). A systematic construction of a dual non-linear bimetric theory of a vortex crystal that respects all global symmetries is an interesting future challenge.

# 5 Conclusions and outlook

In summary, we have explored several mechanisms relevant to the quantum-induced melting of two-dimensional vortex crystals, employing the quadratic low-energy effective Lifshitz theory and its symmetric tensor fractonic dual as our theoretical framework. Using the dual description, we studied the physics of topological defects, and speculated about a direct quantum phase transition from the vortex solid to the vortex liquid phase. We also took initial steps towards a non-linear dual description which we anticipate to be a dynamical theory of gravity.

Finally, we would like to highlight several intriguing research avenues that have emerged from this study of vortex matter in superfluids and warrant further investigation in the future:

- *Vacancy/interstitial RG instability*: An essential task is to gain deeper insights into the nature of the RG instability discussed in Section 2.2 that arises for filling fractions $\nu < \nu_{cr} \approx 8.2$ due to the Andreev-Lifshitz condensation of vacancies and interstitials.

---

[9]Here we use the dynamical metric $\mathfrak{g}_{ij}$ and and its inverse $\mathfrak{g}^{ij}$ to lower and raise indices.

- *Disclination Higgs transition*: It is important to explore whether the direct and exotic continuous Higgs transition, as discussed in Section 3.3, remains intact in the presence of quantum fluctuations or if it gives way to several more conventional transitions, as suggested in [38, 43].

- *Non-linear gravity theory*: Using the ingredients introduced in Section 4, one should be able to construct a non-linear dynamical gravity theory that is consistent with all global symmetries inherent to the problem. This theory could then be employed to compute the decay rate of the Tkachenko mode which should be compared with the result obtained in Ref. [22]. The coupling of the dynamical metric $\mathfrak{g}_{ij}$ and the background metric $g_{ij}$ needs to be understood, and the complete theory of the bimetric model for the rotating superfluid is awaited to be discovered.

Beyond rotating superfluids, it is well-appreciated that the Lifshitz theory emerges at low energies at the critical Rokhsar-Kivelson point in quantum dimmer models and quantum spin ice [3, 42, 65]. Despite originating from a model with different global symmetries, the dual theory that we considered in this paper might inspire a study of the constrained dynamics of the low-energy excitations at the Rokhsar-Kivelson critical point.

*Note added*: We would like to draw reader's attention to the paper [66] by Yi-Hsien Du, Ho Tat Lam, and Leo Radzihovsky, titled "Quantum vortex lattice via Lifshitz duality", that has some overlap with our work.

## Acknowledgements

The authors thank Eddy Ardonne, Eduardo Fradkin, Carlos Hoyos, Leo Radzihovsky, Dam Thanh Son and Wilhelm Zwerger for discussions and comments. S.M. is supported by Vetenskapsrådet (grant number 2021-03685) and Nordita. The work of D.X.N. is supported, in part, by Grant IBS-R024-D1.

## A  Vertex operators in quantum Lifshitz theory

Here following closely Refs. [3, 42], we review the known facts about vertex operators in quantum Lifshitz model in two spatial dimensions.

Consider a quantum Lifshitz theory of a compact scalar $\varphi \in (0, 2\pi)$. The theory is defined by the (Euclidean) Lagrangian

$$\mathcal{L} = \frac{1}{2}(\partial_\tau \varphi)^2 + \frac{\eta^2}{2}(\nabla^2 \varphi)^2. \tag{A.1}$$

Consider perturbations that preserve the $U(1)$ shift symmetry that acts as $\varphi \rightarrow \varphi + \delta$.[10] In particular, we are interested in operators that create vortex defects of $\varphi$ with integer vorticity $m$

$$\tilde{O}_m(\mathbf{x}) = \exp(i \int d\mathbf{z}\, \alpha(\mathbf{z})\Pi(\mathbf{z})), \tag{A.2}$$

where $\alpha(\mathbf{z}) = m \arg(\mathbf{z} - \mathbf{x})$ and $\Pi(\mathbf{z})$ denotes canonical momentum density that is conjugate to the Lifshitz field $\varphi$. When acting on the ground state, this operator shifts the phase to create a

---

[10]Due to the dynamical exponent $z = 2$, the $U(1)$ symmetry is not broken spontaneously, but only exhibits algebraically-decaying correlators of $U(1)$-charged operators.

singular vortex profile. The (spatial) scaling dimension of $\tilde{O}_m$ is [3,42]

$$\tilde{\Delta}_m = 2\pi\eta m^2. \tag{A.3}$$

In other words, the equal-time correlator is given by the power-law

$$\langle \tilde{O}_m(\mathbf{z})^\dagger \tilde{O}_m(\mathbf{z}') \rangle \sim |\mathbf{z} - \mathbf{z}'|^{-4\pi\eta m^2}. \tag{A.4}$$

Due to dynamical critical exponent $z = 2$, for $\tilde{\Delta}_m < 2 + z = 4$, the vortex operator becomes RG-relevant and destabilizes the Lifshitz scale-invariant fixed point. In particular, if we start with a large $\eta$ and decrease it to the value $\eta_c = 2/(\pi m^2)$, $\tilde{\Delta}_m$ becomes marginal.

One can also consider electric vertex operators of the form

$$O_n = \exp(in\varphi) \tag{A.5}$$

which have the (spatial) scaling dimension [3,42]

$$\Delta_n = \frac{n^2}{8\pi\eta} \tag{A.6}$$

and thus

$$\langle O_n(\mathbf{z})^\dagger O_n(\mathbf{z}') \rangle \sim |\mathbf{z} - \mathbf{z}'|^{-\frac{n^2}{4\pi\eta}}. \tag{A.7}$$

These electric operators violate the $U(1)$ shift symmetry and cannot be added to the Lagrangian of the model.

# B    Tkachenko dispersion

We start from the Euler-Lagrange equations of the dual theory (3.7)

$$\frac{\kappa}{4}\partial_t \underbrace{\left(\partial_t a_{ij} - (\partial_i \partial_j - \frac{1}{2}\delta_{ij}\Delta)a_0\right)}_{e_{ij}} + \frac{\lambda}{2B}(\partial_i \tilde{\partial}_j + \partial_j \tilde{\partial}_i) \underbrace{\frac{1}{2B}(\partial_k \tilde{\partial}_l + \partial_l \tilde{\partial}_k)a_{kl}}_{-b} = 0. \tag{B.1}$$

Now without loss of generality we work in the temporal gauge $a_0 = 0$ and consider a wave propagation along the $x$-direction such that $\partial_y = 0$. Under these conditions, $e_{ij} = \partial_t a_{ij}$ and $b = \partial_x^2 a_{xy}/B$. The above field equations simplify to $\partial_t^2 a_{xx} = \partial_t^2 a_{yy} = 0$ and $\frac{\kappa}{4}\partial_t^2 a_{xy} + \frac{\lambda}{2B^2}\partial_x^4 a_{xy} = 0$. The last equation gives us the quadratic dispersion $\omega^2 = \frac{2C_2\lambda}{B^2}q^4$ of oscillations of the off-diagonal component $a_{xy}$.

# C    From old to new dual tensor gauge theory

The fracton-elasticity duality of vortex crystals was investigated in Ref. [38], where a gauge theory involving symmetric tensor gauge fields (dual to elasticity) intertwined with a $u(1)$ vector gauge theory (dual to superfluidity). Here we demonstrate how one can derive from that construction the dual gauge theory (3.7) investigated in this paper.

We start from the intertwined dual gauge theory considered in Ref [38] [11]

$$\mathcal{L} = \mathcal{L}_g(a_\mu) + \frac{1}{2Bn_0}\epsilon_{ij}\left(B^i + B\epsilon^{ik}a_k\right)\partial_t\left(B^j + B\epsilon^{jl}a_l\right)$$
$$+ \frac{1}{2}C_{ij;kl}^{-1}\left(E^{ij} + B\delta^{ij}a_t\right)\left(E^{kl} + B\delta^{kl}a_t\right) + A_{ij}J^{ij} + A_0\rho + 2\pi a_\mu j_v^\mu, \tag{C.1}$$

---

[11]To be consistent with notation in the main text, here we use conventions for currents and gauge fields that differ from [38].

where the elasticity tensor $C_{ij;kl}$ is given by

$$C_{ij;kl} = 8C_1 P_{ij;kl}^{(0)} + 4C_2 P_{ij;kl}^{(2)} \tag{C.2}$$

with the compression and shear projection operators

$$P_{ij;kl}^{(0)} = \frac{1}{2}\delta_{ij}\delta_{kl},$$
$$P_{ij;kl}^{(2)} = \frac{1}{2}\left(\delta_{ik}\delta_{jl} + \delta_{il}\delta_{jk} - \delta_{ij}\delta_{kl}\right). \tag{C.3}$$

The definitions of the electric field and magnetic field in terms of the symmetric tensor gauge field are $E_{ij} = \partial_t A_{ij} - \partial_i\partial_j A_0$ and $B^i = -\epsilon_{jk}\partial^j A^{ki} = \tilde{\partial}_k A^{ki}$. The superfluid current in the dual picture is $j^\mu = \epsilon^{\mu\nu\rho}\partial_\nu a_\rho$, while $n_0$ is the superfluid background density. The electric field $E_{ij}$ and magnetic field $B^i$ are invariant under the gauge transformations

$$A_{ij} \to A_{ij} + \partial_i\partial_j\beta, \quad A_0 \to A_0 + \partial_t\beta. \tag{C.4}$$

The gauge theory is also invariant under the additional gauge transformation

$$A_{ij} \to A_{ij} + \frac{1}{B}\delta_{ij}\xi, \quad a_\mu \to a_\mu - \partial_\mu\xi. \tag{C.5}$$

Notice that the gauge transformation (C.4) differs from the gauge transformation (3.6) since $A_{ij}$ is not traceless. The gauge transformations imply the conservation laws [38] for vortex crystal topological defects (represented by the disclination density $\rho$ and the dislocation current $J^{ij}$) and vacancies/interstitials (represented by their current $j_v^\mu$)

$$\partial_t\rho - \partial_i\partial_j J^{ij} = 0, \tag{C.6}$$

$$\frac{1}{B}J^{ij}\delta_{ij} - 2\pi\partial_\mu j_v^\mu = 0. \tag{C.7}$$

Combining the above equations gives us

$$\partial_t\rho - (\partial_i\partial_j - \frac{1}{2}\delta_{ij}\triangle)J^{ij} - \frac{\pi}{B}\triangle(\partial_\mu j_v^\mu) = 0. \tag{C.8}$$

We observe that in the presence of vacancies, the conservation of the trace of the quadrupole in the main text needs to be modified to $d\tilde{Q}^{tr}/dt = 0$ with

$$\tilde{Q}^{tr} = \int d^2x \left(\mathbf{x}^2\rho - \frac{4\pi}{B}j_v^t\right) \tag{C.9}$$

which implies the modified glide constraint derived in [38]. Note that a similar conclusion for an ordinary crystal was derived in [28].

From now on, we ignore the vacancy current and lattice topological defects and derive from Eq. (C.1) the traceless tensor gauge theory in the main text. At leading order in derivative expansion, the first term in (C.1) does not depend on $a_t$, so the resulting Gauss law

$$B^2\delta^{ij}C_{ij;kl}^{-1}\delta^{kl}a_t + BE^{ij}C_{ij;kl}^{-1}\delta^{kl} = 0. \tag{C.10}$$

Thus $a_t$ is nothing but the trace of the symmetric tensor electric field

$$a_t = -\frac{1}{2B}E^{ij}\delta_{ij}, \tag{C.11}$$

where we used that $P^{(2)}_{ij;kl}\delta^{kl} = 0$. Substituting it now back into the Lagrangian (C.1), we use the explicit form of the projectors and find

$$\mathcal{L} = \mathcal{L}_g(a_\mu) + \frac{1}{2Bn_0}\epsilon_{ij}\left(B^i + B\epsilon^{ik}a_k\right)\partial_t\left(B^j + B\epsilon^{jl}a_l\right) + \frac{\kappa}{8}e^{ij}e^{kl}, \tag{C.12}$$

where we defined the symmetric traceless tensor $e^{ij} = E^{ij} - \delta^{ij}\frac{1}{2}E^{ab}\delta_{ab}$ and $\kappa = C_2^{-1}$. Restricting to the leading-order superfluid Lagrangian $\mathcal{L}_g(a_\mu) = -\lambda(\epsilon^{ij}\partial_i a_j)^2/2$, the equation of motion for $a_i$ is

$$\lambda\tilde{\partial}_i\tilde{\partial}_j a_j + \frac{1}{n_0}\partial_t(B^i + B\epsilon^{ij}a_j) = 0 \tag{C.13}$$

which approximately can be solved by

$$a_i = \frac{1}{B}\epsilon_{ij}B_j + \dots \tag{C.14}$$

Substituting this solution into Eq. (C.12), we finally get

$$\mathcal{L} = -\frac{\lambda}{2}b^2 + \frac{\kappa}{8}e_{ij}e^{ij}, \tag{C.15}$$

where we introduced $b = -\frac{1}{B}\partial_k B_k = -\frac{1}{B}\partial_k\tilde{\partial}_b A^{bk}$. This is exactly our new symmetric traceless gauge theory (3.7).

Finally, if we define the traceless symmetric tensor gauge field $a_{ij}$ and rename $A_0$

$$a_{ij} = A_{ij} - \frac{\delta_{ij}}{2}A_{kk}, \quad A_0 \to a_0 \tag{C.16}$$

we can write the $e_{ij}$ and $b$ in terms of the new gauge fields

$$b = -\frac{1}{2B}(\partial_i\tilde{\partial}_j + \partial_j\tilde{\partial}_i)a_{ij}, \tag{C.17}$$

$$e_{ij} = \partial_t a_{ij} - (\partial_i\partial_j - \frac{1}{2}\delta_{ij}\Delta)a_0. \tag{C.18}$$

After the redefinitions, the gauge transformation of the traceless symmetric tensor gauge field inherited from (C.4) reads

$$a_0 \to a_0 + \partial_t\beta, \qquad a_{ij} \to a_{ij} + (\partial_i\partial_j - \frac{1}{2}\delta_{ij}\Delta)\beta \tag{C.19}$$

which reproduces gauge transformations (3.6) from the main text.

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
