# Peer review of "On quantum melting of superfluid vortex crystals: from Lifshitz scalar to dual gravity"

_SciPost Physics_

## Round 2 · Referee Report · Anonymous (Referee 1) · 2024-4-22

Report

The authors aim to provide a low-energy description of vortex crystals. The paper is well-written, with a clear objective. However, I have several concerns that, if addressed, could further improve the clarity and readability of the paper:
1. The connection between gauge symmetry and glide constraints is not clearly justified. Typically, glide constraints are associated with the requirement to prevent volume changes. In a systematic low-energy theory, I would expect the displacement field to introduce a massive Goldstone mode for conformal transformations, which is subsequently eliminated by a constraint ensuring quadrupole conservation. It is unclear why this would be linked to gauge symmetry.
2. The advantage of replacing the symmetric tensor gauge field with the metric is not apparent.
3. The physical significance of the velocity field remains opaque. The same is true for the manifold described by the metric constructed out of $A_{ij}$ and the diffeomorphism invariance that follows.
4. I do not see why I should use the metric to raise indices. One does not use the field $A_{ij}$ for that purpose.
5. Do defects play any role in the curvature of the Riemann tensor introduced in Section 4?
6. The omission of torsion in the discussion is puzzling.
7. The authors mention the bimetric theory of gravity, presumably drawing on their experience with the Quantum Hall Effect. However, any viscoelastic gapless medium inherently introduces an additional metric due to elastic degrees of freedom. It is uncertain whether the authors would refer to this theoretical approach as 'bimetric' in a general sense or if 'bimetric theory' has a more specific, technical definition that is not addressed in the paper.
8. The prevailing view is that the strain tensor in elasticity couples to the metric as a Higgs field breaking translations on curved manifolds. This is a modern realization of Kleinert ideas, known in the literature. However, I find it difficult to see how the methods introduced in this paper relate to these established models of embedding crystals into curved manifolds.

Recommendation

Ask for major revision

  • validity: high
  • significance: high
  • originality: good
  • clarity: high
  • formatting: perfect
  • grammar: excellent

Author:  Sergej Moroz  on 2024-10-11  [id 4860]

(in reply to Report 1 on 2024-04-22)

The authors aim to provide a low-energy description of vortex crystals. The paper is well-written, with a clear objective. However, I have several concerns that, if addressed, could further improve the clarity and readability of the paper:

-- We thank the Referee for constructive criticism and suggestions which motivated us to improve our paper. In the following, we answer Refere’s comments point by point:

C1: The connection between gauge symmetry and glide constraints is not clearly justified. Typically, glide constraints are associated with the requirement to prevent volume changes. In a systematic low-energy theory, I would expect the displacement field to introduce a massive Goldstone mode for conformal transformations, which is subsequently eliminated by a constraint ensuring quadrupole conservation. It is unclear why this would be linked to gauge symmetry.
A1: As the Referee noted, the original glide constraint is directly related to the conservation of the quadrupole trace. When both vacancies and interstitials are excluded, the quadrupole trace remains conserved, leading to the glide constraint. In our manuscript, we discussed how the traceless symmetric tensor gauge symmetry in the dual description of the vortex lattice incorporates this conservation. The absence of vacancies and interstitials is enforced through the dual relation of the displacement field, $u^i = \epsilon^{ij} \partial_j \phi / B$, along with the condition $\epsilon^{ij} \partial_i \partial_j \phi = 0$.
The relaxation of the glide constraint, referred to as the generalized glide constraint, was thoroughly explored in our previous publication (Ref[38]), where we explicitly derived a dual description from an effective elasticity theory of the displacement field. In Appendix C of the current manuscript, we also establish the connection between the dual descriptions from Ref[38] and our present work.
Within our construction of the Lifshitz duality, the glide constraints (3.14) and (3.19), in the absence and presence of vortex vacancies, respectively, follow directly from the explicit form of the tensor current $j^{ij}$ written in Eq. (3.10).
We promoted a footnote (previously before Eq. (3.14)) to the main text to make our presentation more clear.

C2: The advantage of replacing the symmetric tensor gauge field with the metric is not apparent.
A2: The formulation in terms of the dynamical metric will allow us to describe the physics of the vortex lattice in terms of an emergent dynamical non-relativistic gravity valid in the lowest Landau level approximation. There are several advantages of this approach: First, using the well-developed Newton-Cartan formalism, we will be able to incorporate consistently non-linear terms in the effective gravity model. Moreover, these ideas will provide some additional insight on how to couple the low-energy dynamics of the vortex lattice to background geometry, see also our reply to Referee 2.
To convey these arguments, we modified the conclusion section of our revised manuscript appropriately.

C3: The physical significance of the velocity field remains opaque. The same is true for the manifold described by the metric constructed out of Aij and the diffeomorphism invariance that follows.
A3: The diffeomorphism transformation (4.8) implies that under a Galilean boost $x^i \to x^i-\alpha^i t$, the field $v^i$ transforms by a constant shift, which is what is expected for a velocity. The volume-preserving diffeomorphism is a non-linear generalization of the u(1) gauge redundancy of the dual to the Lifshitz theory in Sec. 3. The volume-preserving diffeomorphism algebra also has the physical meaning of the long-wave-length limit of the Girvin MacDonald Platzman (GMP) algebra of the lowest Landau level limit.
We modified Sec. 4 appropriately to incorporate these comments.

C4: I do not see why I should use the metric to raise indices. One does not use the field Aij for that purpose.
A4: In the linearized theory, one can use the flat metric ($\mathcal{g}^{ij}=\delta^{ij}$) to raise the indices. However, in order to construct systematically the non-linear effective theory defined on a flat background surface, in which the non-linear area-preserving diffeomorphism in Eq. (4.4) is a symmetry, one should use the emergent metric $\mathcal{g}^{ij}$ to raise the indices of $\mathcal{s}_{ij}$ to construct the invariant term $\mathcal{s}_{ij}\mathcal{s}^{ij}$. The effective field theory will be invariant under the emergent non-linear area-preserving diffeomorphism if the action is a scalar with all the indices contracted and raising and lowering indices using the dynamical metric tensor.
We updated Sec. 4 to clarify this point.

C5: Do defects play any role in the curvature of the Riemann tensor introduced in Section 4?
A5: From Eq. (4.6) it becomes clear that the Ricci tensor represents a coarse-grained superfluid density fluctuation on top of the ground state. This in turn is affected by the presence of defects. Indeed, combining the Gauss law and the Ampere law in Eq. (3.25) and Eq. (3.26). and the mapping to the gravitational model in Sec 4, one can see that disclinations and dislocations are sources for the emergent geometry which determines the Riemann curvature. See also our answer to point 6 below.

C6: The omission of torsion in the discussion is puzzling.
A6: Notice that our emergent gravity theory originates from the *dual* u(1) gauge theory and thus is very different to the well-established geometric theory of defects, described for example in [arXiv:cond-mat/0407469]. In the latter theory the emergent metric originates directly from the strain tensor (i.e. the non-dual variable) and the emergent curvature and torsion tensors are simply related to densities of topological defects. The two geometric descriptions are very different since they are related by a non-trivial duality transformation. To resolve Referee's concern, we added a new footnote 8 in our manuscript.

C7: The authors mention the bimetric theory of gravity, presumably drawing on their experience with the Quantum Hall Effect. However, any viscoelastic gapless medium inherently introduces an additional metric due to elastic degrees of freedom. It is uncertain whether the authors would refer to this theoretical approach as 'bimetric' in a general sense or if 'bimetric theory' has a more specific, technical definition that is not addressed in the paper.
A7: As we mentioned in the previous point, we believe that the elastic dynamical metric that the Referee has in mind is related to our dynamical metric by a duality transformation, so these two dynamical metrics should not appear together in the bimetric theory. Instead in the to-be-constructed bimetric theory, the dynamical emergent metric introduced in Section 4 will couple to the background non-dynamical metric $g_{ij}$ that is determined by a surface on which the rotating Bose-Einstein condensate resides. Due to similar lowest Landau level constraints shared by fractional quantum Hall fluids and vortex lattices, we expect that the FQH bimetric theory developed in Ref [61] will provide useful guidance for the bimetric model of the vortex lattice.
We extended our discussion about the bimetric theory in Sec. 4.

C8: The prevailing view is that the strain tensor in elasticity couples to the metric as a Higgs field breaking translations on curved manifolds. This is a modern realization of Kleinert ideas, known in the literature. However, I find it difficult to see how the methods introduced in this paper relate to these established models of embedding crystals into curved manifolds.
A8:Building upon our answers to points 6 and 7 above, the dynamical metric we are introducing here is related to a more established strain dynamical metric by a non-trivial duality transformation. Therefore, it is not surprising that our dynamical metric couples differently to the background surface geometry as compared to the strain dynamical metric. The development of the bimetric theory in our dual formulation is an exciting future direction that goes beyond the scope of this paper.

---

## Round 2 · Referee Report · Anonymous (Referee 2) · 2024-5-22

Report

This paper uses the fracton/elasticity duality to elucidate the melting process of vortex lattices. The starting point is the effective action proposed in [20] which then is dualized into a fractonic theory with Hubbard-Stratonovich fields $b$ $e^{ij}$. The new theory correctly captures the quadratic dispersion of the so-called Tkachenko mode. In their construction, the dual description of the vortex crystal defects is in terms of fractonic matter. In addition, the authors explore the slightly disconnected question of embedding the fracton gauge symmetry obtained into the non-linear volume-preserving diffeomorphism group.

The paper is scientifically sound and gives insight into the active research topics of the melting of vortex lattices and into the puzzling relation between fractons and gravity. Nonetheless, I have a main criticism/question that I would appreciate if the authors could clarify before I recommend the paper for publication.

Following the construction of [53] the authors embedded the theory's gauge transformations into the volume-preserving group, however, their gauge theory has been built after assuming several approximations such that their model captures only the gapless mode of the vortex crystal. Therefore, it is not obvious that the non-linear gravity theory Eq. 4.9 will correspond with the low energy description of a larger model capturing the dynamics of the gap mode (Kohn mode). If that is not the case, it is not clear the physical relevance of such embedding.

Recommendation

Ask for minor revision

  • validity: -
  • significance: -
  • originality: -
  • clarity: -
  • formatting: -
  • grammar: -

Author:  Sergej Moroz  on 2024-10-11  [id 4861]

(in reply to Report 2 on 2024-05-22)

We thank the Referee for their assessment and criticism which motivated us to improve our paper.

Here we address the main question asked by the Referee about the dual gravity construction: Indeed, our approach closely follows Ref [53] and therefore we work in the lowest Landau level limit. In this approximation, only excitations with frequencies much lower than the cyclotron frequency (which fixes the energy gap of the Kohn mode) are incorporated. So the Referee is indeed correct that our dual gravity model will not capture the physics at the scale of the high-energy Kohn mode which originates from a higher Landau level. The key purpose of the geometric formulation is therefore to embed the low-energy dynamics of the vortex lattice into a non-relativistic Newton-Cartan gravity model, where the Tkachenko wave can be identified as an emergent massless dynamical graviton excitation. By imposing all global symmetries, in the future we would like to systematically construct the gravity model and extract leading non-linear terms that determine the scattering and decay rate of the Tkachenko wave in the rotating vortex crystal. We can further formulate the topological defects of the vortex lattice as the matter field in the gravitational model (following Ref[60]).

Furthermore, inspired by the bimetric formalism developed for fractional quantum Hall fluids in Ref[61], we expect that the gravity formulation will help us to construct consistently the coupling of the vortex lattice with a background surface metric, which in turn will help us to analyze the physical responses such as viscosity and thermal transport.

So, the formulation of the vortex lattice as an effective theory of gravity initiated in our paper paves the way to study various hard problems using the well-developed Newton-Cartan formalism.

To convey these arguments, we modified the conclusion section of our revised manuscript appropriately.

---

## Editorial Decision

resubmitted